# Continuous Culture of *Auxenochlorella protothecoides* on Biodiesel Derived Glycerol under Mixotrophic and Heterotrophic Conditions: Growth Parameters and Biochemical Composition

**DOI:** 10.3390/microorganisms10030541

**Published:** 2022-02-28

**Authors:** Evagelina Korozi, Vasiliki Tsagou, Io Kefalogianni, Giorgos Markou, Dimitris Antonopoulos, Lambis Chakalis, Yannis Kotzamanis, Iordanis Chatzipavlidis

**Affiliations:** 1Laboratory of General and Agricultural Microbiology, Department of Crop Science, Agricultural University of Athens, Iera Odos 75, 11855 Athens, Greece; evangeliakorozi@yahoo.com (E.K.); bmic7kei@aua.gr (I.K.); 2Laboratory of Food Biotechnology and Recycling of Agricultural By-Products, Institute of Technology of Agricultural Products, Hellenic Agricultural Organization-Demeter, Leof. Sofokli Venizelou 1, Lykovrysi, 14123 Athens, Greece; gmarkou@itap.com.gr; 3Pavlos N. Pettas SA, Industrial Area of Patras Block 28, B7, 25200 Patras, Greece; dimitris.antonopoulos@pnpettas.gr (D.A.); lambis.chakalis@pnpettas.gr (L.C.); 4Fish Nutrition Laboratory, Hellenic Centre for Marine Research, Institute of Marine Biology, Biotechnology and Aquaculture, Agios Kosmas, Hellenikon, 16777 Athens, Greece; jokotz@hcmr.gr

**Keywords:** *Auxenochlorella protothecoides*, heterotrophy, mixotrophy, crude glycerol, continuous fermentation, growth kinetics, fatty acids profile, amino acids profile

## Abstract

As crude glycerol comprises a potential substrate for microalga fermentation and value added products’ biosynthesis, *Auxenochlorella protothecoides* was grown on it under heterotrophic and mixotrophic conditions and its growth kinetics were evaluated in a continuous system under steady state conditions. Increasing initial glycerol concentration (from 30 to 50 g/L) in the heterotrophic culture led to reduced biomass yield (*Y_x/S_*) and productivity (*P_x_*), but favored lipid accumulation. Under heterotrophic conditions, the microalga was found to grow better (biomass up to 7.888 g/L) and faster (higher growth rates), the system functioned more effectively (higher *P_x_*) and crude glycerol was exploited more efficiently. Heterotrophy also favored proteins synthesis (up to 53%), lipids (up to 9.8%), and carbohydrates (up to 44.6%) accumulation. However, different trophic modes had no significant impact on the consistency of proteins and lipids. Oleic acid was the most abundant fatty acid detected (55–61.2% of the total lipids). The algal biomass contained many essential and non-essential amino acids, especially arginine, glutamic acid, lysine, aspartic acid, leucine, and alanine. In all the experimental trials, the protein contents in the microalgal biomass increased with the increasing dilution rate (*D*), with a concomitant decrease in the lipids and carbohydrates fractions.

## 1. Introduction

Biodiesel as an alternative fuel has attracted increasing attention worldwide in the past few years. During the biodiesel manufacturing process, and, specifically, the production of fatty acids esters by the mix of triacylglycerols with alcohol and catalysts, crude glycerol is produced as a primary byproduct. For every liter of biodiesel produced, around 0.08 kg of crude glycerol is generated [1]. Glycerol is prohibitively expensive to be converted and purified into materials that can be used in the food, cosmetic or pharmaceutical industries due to its high concentration of many impurities, such as soap and methanol. The market has been flooded with crude glycerol over recent years and, thus, seeking new ways to dispose of this waste stream seems to be imperative. Hence, the microbial fermentation of crude glycerol and its conversion into other useful products (such as polyhydroxyalkanoates, docosahexaenoic acid, lipids, etc.) is of special interest [2]. 

Microalgae are fast growing and widely distributed in nature unicellular species, which stand out for their high value biomass. Microalgal biomass serves as a unique platform for biofuels and bioproducts, many of which have several applications in the pharmaceutical, cosmetics, and food industries. Besides traditional photoautotrophic microalgal mass culturing, a surprising number of species are capable of heterotrophic or mixotrophic growth using organic carbon sources, such as glucose, glycerol, fructose, lactose, and mannose [3]. The production of biomass through heterotrophy or mixotrophy provides an opportunity to improve productivity, but requires inexpensive raw materials to be economically and commercially viable. Some wastes (low cost feedstock) contain organic and inorganic components that may serve as nutrients for algal growth, decreasing the culture media cost and, thus, overall process expenses. This observation makes biodiesel derived glycerol a promising candidate for microalgal growth, due to its high availability and low cost production. 

The selected algal strain *Auxenochlorella protothecoides*, formerly known as *Chlorella protothecoides*, a typical oleaginous and, thus, commercially important green microalga, belongs to the oldest algal strains maintained in culture [4]. *A. protothecoides* was first isolated from the sap of *Populus alba* and displayed morphological similarity with *Chlorella vulgaris*, the type species of the genus, i.e., small spherical cells, reproduction by autospores, and one chloroplast. In contrast to *C. vulgaris, A. protothecoides* demonstrates stronger dependence on organic compounds and the absence of a pyrenoid [4]. *Chlorella* species are robust microorganisms that can grow under various conditions around the world; they can grow, for example, heterotrophically or mixotrophically on glucose, glycerol, acetate, or other organic compounds derived from waste resources, with zero or negative costs in order to accumulate lipids for biodiesel production [5]. 

*A. protothecoides* is able to grow photoautotrophically, mixotrophically, and heterotrophically as well [3]. Specifically, it displays a capability of transition between heterotrophy and photoautotrophy in response to light/dark and the C/N ratio changes, and, hence, it demonstrates multiple metabolic pathways that can be adapted for the biosynthesis of valuable products [5]. *A. protothecoides* has great potential to serve as a source of food and energy, due to its high photosynthetic efficiency (in theory up to 8%) [6], and its ability to accumulate large quantities of neutral lipids (up to 70%) [5] at relatively high densities utilizing a range of fixed carbon sources [7]. In fact, its lipid profile has been reported to display similar contents to vegetable oils and to improve biodiesel properties [8], arousing even more interest in the production of valuable metabolites. There are many nutritional and environmental factors that control the cell growth and its content in value added metabolites, such as carbon source, nitrogen source, essential macro- and micronutrients such as magnesium and copper, temperature, agitation speed, etc. [5]. The exploration of the potential mechanism of value added product biosynthesis, the effect of nutrients on microalgal biomass production, and the recording of physiological and biochemical changes during organic carbon fermentation, have recently attracted scientific interest. To study microalgal characteristics, it is essential to analyze the biochemical composition and understand the transformation pathways of the components in the microalgal cells, which ultimately depend on nutrient availability and their concentrations in the growth medium [5,8]. Furthermore, to define the actual metabolite production capacity of a strain, except for evaluating their content in the biomass, care should be taken to correctly evaluate the metabolites productivities as well [9].

Although the growth of *Auxenochlorella* sp. has been extensively studied, there is still little knowledge about the impacts of glycerol concentrations and different trophic modes on microalgal physiological and biochemical characteristics under steady state conditions. Continuous culture systems are not only generally more representative than batch cultures because of the dynamic equilibrium between nutrient influent and cell metabolism, but they are usually also more productive due to the reduced process downtime required for cleaning, sterilization, and setup. In addition, since they operate in steady state conditions, the control of the whole process can be simpler and more accurate [10,11]. Most studies on continuous microalgae cultivation have been conducted under autotrophic conditions, whereas only a few focus on heterotrophic continuous microalgal cultures and even fewer have been carried out under mixotrophic conditions [9,10,11].

The aim of the present study was to explore the extent to which mixotrophic *A. protothecoides* can be influenced by the presence of light in upcycling the carbon substrate from the culture medium and to examine whether heterotrophic *A. protothecoides* biosynthetic activities can be influenced by the initial glycerol concentration under steady state conditions. To achieve these goals, growth parameters, such as system productivity, biomass yield on glycerol, and the biomass biochemical composition of *A. protothecoides* were estimated in different glycerol concentrations under both heterotrophic and mixotrophic conditions under the steady state situations of a continuous culture system. Under steady state conditions, the measurement of these parameters is reliable and concise, as the biomass productivity and its composition are stable [9]. Nevertheless, such a situation does not occur under batch culture conditions, in which both growth rate and biomass composition change as growth progresses, and, although the maximum metabolites content values are displayed, no accuracy should be expected regarding metabolite productivity calculated at this point [12].

## 2. Materials and Methods

### 2.1. Algal Strain, Medium and Subculture Conditions

The algal species *A. protothecoides* (CCAP 211/8D) from the Culture Collection of Algae and Protozoa SAMS Limited Scottish Marine Institute, Dunderg, Oban, Scotland, United Kingdom was used. Modified-BBM (with the same composition as BBM except for the 3-fold concentration of NaNO_3_ and the addition of vitamins) was used as the basal growth medium supplemented with glucose (10 g/L) and peptone (2 g/L) [13]. Cells were maintained in 250 mL Erlenmeyer flasks each containing 150 mL of the medium and incubated mixotrophically (under 18/6 h of light/dark regimen) and axenically, at 25 °C, on an orbital shaker set to 150 rpm. These cultures were used for the inoculation of the reactors and were subcultured axenically every 3 weeks for maintenance.

### 2.2. Continuous Culture Conditions

Continuous cultures were performed in a 1.5 L New Brunswick Bioflo C30 fermenter, (New Brunswick Co. Inc., Edison, NJ, USA) with a working volume of 350 mL. Agitation was provided by four turbine impellers and was maintained during the cultivation at 150 rpm. The cultures were aerated with 0.2 μm filter sterilized air, while the incubation temperature was T = 25 ± 1 °C. All cultures were axenic and performed under sterile conditions. Crude glycerol in the fresh medium was applied at two different concentrations (30 and 50 g/L). Microalgal growth was evaluated under both heterotrophic (dark conditions) and mixotrophic conditions (illumination by a red LED light–SMD type, 14.4 W per meter, 60 SMD LEDs per meter, photoperiod set at 16 h light and 8 h dark and light intensity of 3000 lux). A red light monochromatic source was chosen on the basis of preliminary trials (data not shown) and the results reported in previous work [14], according to which red light results in high growth rates of *A. protothecoides* under mixotrophic conditions.

Preliminary trials of *A. protothecoides* clarified that peptone, compared to other nitrogen sources, promotes microalgal growth (nitrates and ammonium nitrogen could not be uptaken) and, thus, it was used as the nitrogen source in the present study, at concentrations corresponding to a C/N ratio equal to 20:1. The continuous cultures were inoculated with 10% (vv^−1^) of exponentially grown inocula and were left to operate in batch culture mode for four days, before a continuous flow of feed medium commencing at various dilution rates (*D*). Steady state conditions were obtained after a continuous flow of at least five working volumes of the culture medium. Five mL samples were then taken at daily intervals and analyzed for microbial mass and remaining glycerol concentration until constant values, in order to verify the steady state achievement. It should be noted that the photosynthesis contribution under heterotrophic conditions was considered negligible, since an opaque blanket was wrapped around the glass vessel.

### 2.3. Analyses

Cells were harvested from the growth medium through centrifugation (13,000 rpm for 10 min), washed twice with distilled water and dried overnight at 80 °C until constant weight. Biomass concentration was also indirectly measured by the optical density (OD) at the wavelength of 750 nm [15].

The residual glycerol content in the supernatant was estimated using a glycerol assay procedure kit by Megazyme (Megazyme, Bray, County Wicklow, Ireland).

For carbohydrate content determination a modified phenol-sulfuric acid method was used [16]: in brief, in 0.5 mL of cell sample containing 10–50 mg/L carbohydrates, 10 μL of 90% phenol solution were added and mixed, followed by the addition of 1.25 mL of concentrated sulfuric acid (96%). After 30 min, OD was measured at 485 nm using D-glucose as the standard sugar.

Cellular lipids were extracted with 2:1:0.2 chloroform:methanol and water and were estimated according to a modification of the sulfo–vanillin method [17]. In brief, 20 μL of the cell samples, which contained lipids within the range of 200–500 mg/L, were incubated in a water bath at 80 °C to evaporate chloroform. Thereafter, 0.4 mL of 96% sulfuric acid were added and the samples were placed in a boiling water bath for 10 min. After cooling at room temperature, 1.0 mL of phosphoric acid/vanillin solution was added (for the solution stock 0.12 g of vanillin were firstly dissolved in 20 mL DI water and then in 80 mL of 85% phosphoric acid). Next, the samples were incubated at 37 °C for 15 min. Finally, OD was measured at 530 nm using canola oil for the standard curve.

Proteins were extracted with 0.5 N NaOH and assayed according to Lowry et al. [18] as follows: 1.5 mL of the samples was firstly centrifuged and the obtained pellet was resuspended in 1.5 mL of 0.5 N NaOH and incubated at 100 °C for 20 min. A volume of 100 μL of the extracted proteins was mixed with equal quantity of SDS (5%), followed by the addition of 1 mL of a solution of 2% Na_2_CO_3_ in 0.1N NaOH and then left for 15 min at room temperature. Next, 100 μL of freshly prepared 1 N Folin & Ciocalteu reagent (Penta, Chemicals Unlimited, Prague, Czech Republic) were added. After 30 min in dark, OD was measured at 750 nm. Bovine serum albumin was used as a protein standard.

The fatty acid profile was evaluated according to the AOCS Official Method Ce 1c-89, while the amino acid composition of the algal biomass was analyzed after acid hydrolysis (6 N, 110 °C, 24 h) and derivatization by AccQ-Tag^TM^ Ultra according to the amino acid analysis application solution (Waters Corporation, Milford, MA, USA), as described by Kotzamanis et al. [19].

### 2.4. Calculations

Growth and nutrient dynamics in the chemostat were investigated based on Monod’s model [20], which considers that substrate concentration is the determinant of cellular growth:(1)$$\mu =\mu_{max}\;\frac{S}{S+K_{S}}$$
where, *S* is the concentration of the limiting nutrient in the chemostat which is the determinant of specific growth rate (*μ*) and the maximum specific growth rate (*μ_max_*) as well, and *K_S_* is the saturation constant, which is the half velocity constant denoting the concentration of substrate *S* that supports a rate equal to *μ* = *μ_max_*/2. In other words, *K_S_* is a kinetic parameter that indicates how fast the maximum specific growth rate is being reached.

Biomass productivity (*P_x_*) is the relation of the variation of biomass concentration in time. In continuous processes where the growth is kept exponential at a specific growth rate equal to dilution rate, biomass productivity is expressed as:(2)$$P_{x}=\mu \;\cdot \;x=D\;\cdot \;x$$
where *x* is the dry weight of biomass.

The productivities (*P_m_*) for all the produced metabolites (lipids, proteins, or carbohydrates) were calculated as:(3)$$P_{m}=D\;\cdot \;m$$
where *m* is the dry weight of the metabolite.

## 3. Results and Discussion

### 3.1. Growth Dynamic Parameters under Different Trophic Modes and Feed Glycerol Concentrations

To investigate the growth and the response of *A. protothecoides* to different trophic modes (heterotrophy and mixotrophy) and feed glycerol concentrations, a series of continuous cultures were carried out at various *D*. A first experimental run was conducted in the absence of light, at a feed glycerol concentration of 30 g/L and peptone at a C/N ratio of 20/1. Complex nitrogen sources are presumed to be superior to simpler nitrogen sources (such as nitrate or urea) for heterotrophic microalgal growth, since they also provide amino acids, vitamins, and growth factors [21,22,23]. This continuous culture was performed within the range of *D* = 0.064–0.11 h^−1^ (Figure 1, Table 1). Dry biomass concentrations and steady state yield coefficient on glycerol (*Y_x/S_*) were higher at the lower *D*, with a maximum value of biomass concentration equal to 5.34 g/L. Increasing *D* from 0.08 to 0.95 h^−1^ caused biomass decrease (up to 1.8–0.95 g/L), whereas a further increase in *D* (above 0.12 h^−1^) showed that growth at these *D* was close to the washout conditions.

Within the dilution rates applied, the residual glycerol concentration increased with the increasing dilution rate. Applying Monod’s kinetic model and plotting the 1/*D* vs. 1/*S* curve, the *μ_max_* and *K_S_* values were determined as 0.134 h^−1^ and 3.14 g/L, respectively. The *μ_max_* determined agreed with the fact that the cells were washed out when *D* increased from 0.120 to 1.155 h^−1^. The maximum biomass production of the system (*P_max_*) was estimated equal to 0.393 g/L·h at *D* = 0087 h^−1^. 

Afterwards, the physiological responses of *A. protothecoides* with the increase in feed glycerol concentration (50 g/L) were investigated in a continuous system at different dilution rates within the range of 0.027–0.16 h^−1^, under heterotrophic conditions (Table 2, Figure 2). The trend of biomass production with *D* was the same as in the former experimental case, i.e., it was higher at the lower *D* (≤0.067 h^−1^) with the highest value of 7.88 g/L recorded at *D* = 0.027 h^−1^. The *P_max_* was determined as equal to 0.584 g/L·h at *D* = 0.101 h^−1^, while the estimated *μ_max_* and *K_S_* values were 0.174 h^−1^ and 9.15 g/L, respectively. Although *A. protothecoides* grew with a *μ_max_* almost the same as that estimated in 30 g/L crude glycerol, the *K_S_* value was significantly higher, indicating the lower substrate affinity of the microalga under these operating conditions. In the present study, biomass production was found to be, in general, reduced at higher dilution rates at both 30 and 50 g/L initial glycerol concentrations. Similar behavior has been observed in the literature for several microorganisms, such as *Schizochytrium limacinum* [24] and the yeast *Pichia angusta* [25].

Even though higher biomass concentrations were produced when *A. protothecoides* was grown in the increased initial glycerol concentration, *Y_x/S_* and *P_x_* values at this experimental run were significantly lower compared to the relative ones in the case of feed glycerol concentration of 30 g/L, indicating that increasing glycerol concentration did not contribute to biomass productivity. This finding, in combination with the observation that by increasing the feed glycerol concentration higher amounts of unconsumed glycerol were detected in the growth medium, leads to the assumption not only that glycerol in higher initial concentrations cannot be adequately catabolized, but also that it probably causes inhibition of carbon uptake and microalgal biomass production, acting as a growth restrictive factor. The case of cell growth being negatively influenced at higher substrate concentrations has also been reported for *A. protothecoides* grown in a continuous system with different feed concentrations of glucose [5]. Besides, the phenomenon of an increased residual glycerol concentration with the increasing dilution rate and the increasing feed substrate levels was also observed in the continuous culture of other microorganisms. For example, growing continuously *S. limacinum* at higher dilution rates and feeding glycerol equal to 30 g/L resulted in the existence of certain amounts of residual glycerol in the reactor [20]. Similarly, when *Nitzschia laevis* was grown at *D* > 0.3 day^−1^ or higher feed glucose concentration (>20 g/L), the steady state residual glucose increased [26]. Xiong et al. [27] suggested that continuous feeding of low glucose concentrations is probably the right strategy for *A. protothecoides* culture in a bioreactor. Of particular interest are the results of Zhang and his associates [28], who also studied *C. protothecoides* grown heterotrophically in a fermenter using glucose as the carbon source and urea as the nitrogen source, and employed a new growth model, since, according to them, the Monod model often fails to account for the substrate inhibition of growth at higher substrate concentrations. 

To clarify the growth response of *A. protothecoides* in the presence of light, it was subsequently cultivated in a continuous culture with crude glycerol under a monochromatic source of light. Utilizing artificial light for microalgal growth is of great interest for the production of high value products, such as fatty acids, proteins, and pigments. Light emitting diodes (LEDs) can serve efficiently as a light source for microalgal growth compared to traditional lighting sources and, as they emit only at given bands of wavelength, can provide ways to control and manipulate the biochemical composition of the microalgae [29,30]. A monochromatic red light source (3000 lux) with a photoperiod of 16 h light and 8 h dark was chosen for the mixotrophic continuous culture of *A. protothecoides*. This experimental run was carried out using an initial concentration of 30 g/L glycerol and peptone at a C/N ratio of 20/1 (Figure 3, Table 3).

Within the dilution rates investigated (0.27 to 0.107 h^−1^), biomass production was higher at the lower *D*, getting a maximum value of 3.44 g/L and decreased significantly with increasing *D* above this value, leading to the culture’s wash out. The higher values of *Y_x/S_* and *P_x_* at *D* ≤ 0.08 h^−1^ indicated more efficient glycerol assimilation and a more productive system function at these *D* under the specific conditions. The maximum specific growth rate (*μ_max_*) for this experimental run was estimated equal to 0.129 h^−1^, while the maximum productivity of the system (0.153 g/L·h) was achieved at *D* = 0.06 h^−1^. The saturation constant was the highest of all the experimental cases, equal to 10.64 g/L.

Nevertheless, both *Y_x/S_* and *P_x_* exhibited lower values at all the *D* applied when *A. protothecoides* was cultivated mixotrophically than under heterotrophic conditions, regardless of the feed glycerol concentration. Under mixotrophic conditions, the remaining glycerol concentration in the medium was quite high (≥10%), even at the lower *D* applied. Consequently, by comparing the growth parameters of *A. protothecoides* grown in continuous cultures with a feed glycerol concentration of 30 g/L, under either heterotrophic or mixotrophic conditions, we concluded that, under the heterotrophic mode, firstly, the microalga could grow faster (higher *μ_max_* values); secondly, the continuous system could function more effectively (higher *P_max_* values); and finally, it could exploit crude glycerol more efficiently (lower *K_S_* values). These results confirm the hypothesis of our previous work conducted in batch cultures [30], that *A. protothecoides* growth is not favored under mixotrophic conditions, in opposition to what has been stated for other microalgae. Light seems to have a potential inhibitory effect on *A. protothecoides* when grown mixotrophically, suggesting that *A. protothecoides* is not favored by the mixotrophic conditions, probably due to a combination of reduced organic carbon assimilation and lower photosynthetic performance. 

Inhibition of the substrate assimilation in the presence of light has been also reported from Xiao et al. [31], when *A. protothecoides* was cultivated mixotrophically in a continuous system utilizing glucose instead of glycerol. In that case, glucose metabolism was also found to be restricted from the presence of light leading to limited biomass and lipids production. According to Heredia-Arroyo et al. [5], *A. protothecoides* did not seem to be able to utilize light in the presence of glucose, like an amphitrophic organism, as the growth stimulating effects of light and CO_2_ utilization in mixotrophic cultures were not as strong as the effects of glucose. Other researchers have reported that the availability of light decreases the microalgae mixotrophic capability of exploiting organic substrates. Regarding *C. vulgaris*, Haass and Tanner [32] described an inducible hexose transport system with glucose as an inducer that could be inhibited in the presence of light. Apparently, when light is supplied to the culture, microalgae do not efficiently consume the organic matter, while growth is also limited by other nutrients as well [33]. On the other hand, there are many reports claiming that, during mixotrophic cultivation, several microalgae strains, such as *Phaeodactylum tricomutum,* and *Chlorococcum sp*., resulted in higher biomass production as a combination of photosynthesis and organic carbon consumption [34,35]. Taking all the above into consideration leads us to the conclusion that the *A. protothecoides* phenotype of limited growth and restricted consumption of organic carbon in the presence of light differs from those of other microalgae. 

### 3.2. Proteins, Lipids and Carbohydrates Production

#### 3.2.1. Carbohydrates

Among the different organic compounds in microalgal biomass, the most energy rich group are lipids (37.6 kJ/g), followed by proteins (16.7 kJ/g) and carbohydrates (15.7 kJ/g) [36]. Even though carbohydrates represent the lowest energy content, they comprise an important substrate for the production of several significant biofuels (such as bioethanol, biobutanol, biohydrogen, etc.) through biotechnological conversion technologies [37]. Carbohydrates production serves a dual role in the microalgal cell: they are structural components of the cell wall and storage compounds inside the cell, providing the energy for several metabolic processes if needed. However, the majority of studies examining the impact of different trophic modes, substrates and other environmental factors on microalgal growth focus on lipid production and, thus, little is known about the effect of these parameters on carbohydrate accumulation. In the present study, the biomass content in carbohydrates, as well as carbohydrates productivity (*P_c_*), were estimated in the steady state situation under the different experimental conditions applied (Table 1, Table 2 and Table 3, Figure 1, Figure 2 and Figure 3). Different trophic modes, as well as different initial glycerol concentrations, were found to affect the biosynthesis of carbohydrates. Under heterotrophic conditions, the carbohydrate content in microalgal cells was significantly enhanced. Specifically, in the case of 50 g/L feed glycerol concentration, a maximum carbohydrate content of up to 44% of the dry biomass was recorded at *D* = 0.027 h^−1^. However, similar values were observed at the same *D* between the two heterotrophic trials, indicating that increased initial glycerol concentration had no significant impact on the carbohydrates fraction. It seems that an enhanced carbon source did not arouse enhanced substrate assimilation via carbohydrates biosynthesis pathways. Taking into account the rather significant accumulation of carbohydrates that occurred under the certain heterotrophic conditions, in combination with studies with similar results that have been already reported, *Auxenochlorella* could be included among the microalgal species that usually display a high innate carbohydrate content under certain conditions and, thus, may constitute a good feedstock for biofuels production [14,38].

When *A. protothecoides* was cultivated mixotrophically on glycerol, both carbohydrate content and carbohydrate productivity were distinctively decreased. At all *D* applied the relative values were significantly lower than those recorded under heterotrophic conditions. This observation indicates that under the certain mixotrophic conditions *A. protothecoides* probably assimilates glycerol and utilizes it for the synthesis of other energy-consuming compounds like pigments (chlorophyll or carotenoids).

#### 3.2.2. Proteins

Proteins comprise a significant fraction of the biomass of actively growing microalgae. The effects of different trophic modes (heterotrophy, mixotrophy) and different feed glycerol concentrations on *A. protothecoides* protein content (% of dry weight) in a continuous system at different *D* are given in Table 1, Table 2 and Table 3 and Figure 1, Figure 2 and Figure 3. The fact that microalgal protein fraction increased with increasing *D* comes as a general observation of the three experimental trials. This finding can be possibly explained by the fact that nitrogen repletion achieved by higher feeding rates, promotes protein production, while nitrogen depletion (probably existing in the culture fluid at lower feeding rates) reduces protein content but favors starch and/or fat accumulation in cells [39,40]. In general, enhanced contents of proteins were detected in all experimental cases (up to 40%). By increasing the feed glycerol concentration heterotrophically the protein productivity (*P_Pr_*) was decreased. These results were expected considering that proteins are primary metabolites that in general follow the growth pattern. Higher-growth rate cultures tend to produce more protein [41]. However, there is a previous study on *Arthrospira platensis* growing in glycerol, where the microalgal protein content increased with the increase in glycerol concentration, probably due to a somehow metabolic assimilation of glycerol that facilitated the protein synthesis [42]. In the present case, the impact of the two different trophic modes on the protein content was the same as that on the microbial growth. Specifically, between heterotrophy and mixotrophy, the protein fraction was enhanced under heterotrophic conditions, where the microalgal growth was also favored. 

With the aim to define the quality of produced proteins in the microalgae biomass, the amino acid profiles were analyzed (Figure 4). 

In all treatments conducted, the algal cells contained almost all the essential amino acids for human nutrition. During heterotrophic growth at all different *D* applied, arginine (11.7%), which is a conditionally essential amino acid, glutamic acid (12.6%), alanine (8.8%), lysine (8.4%), aspartic acid (8.3%) and leucine (8.5%) were the most abundant amino acids detected. Lower contents of serine, glycine, threonine, proline, valine, isoleucine, and phenylalanine were also recorded, while hydroxyproline, histidine, tyrosine, methionine, and cysteine were found only in traces. Different *D* during the continuous system cultivation did not seem to affect amino acids production with the exception of arginine, the content of which, in the microalgal cells, was measured slightly higher with increasing *D*. Similar results were obtained by the amino acid profiling during the mixotrophic cultivation of *A. protothecoides*, indicating that different trophic mode may affect the total produced proteins amounts, but do not affect the consistency of proteins. Only minor differences were found in the levels of specific amino acids between the two trials. Hence, the amino acids profile of the mixotrophically grown cells followed the heterotrophic pattern. Namely, six amino acids (alanine, aspartic acid, glutamic acid, leucine, lysine, and glycine) were responsible for 50% or more of the total detected amino acid contents in the microalgal cells. An equivalent to our results amino acid profile of *A. protothecoides* cells grown heterotrophically was recorded from Lane et al. [43]. However, according to their results, the different trophic modes (phototrophy and heterotrophy in that case) had a significant impact on both the percentage of each amino acid in the protein fraction and the total protein content (with phototrophy displaying higher values), but this may be accounted for due to the manipulations used under the specific conditions that directed the microalgal metabolism to focus on lipid production instead of protein biosynthesis.

#### 3.2.3. Lipids

Continuous culturing is a more effective approach to investigating the fatty acid content of the algae biomass, compared to batch and fed batch cultures. Specifically, continuous flow cultures provide a way to study lipid production at specific, stable C/N ratios, at constant specific growth rates and biomass concentrations, and give stable fatty acid profiles at fixed operational conditions [23]. In this study, *A. protothecoides* was grown on glycerol in a continuous system and at all the steady states situations achieved the microalgal lipid content and the lipid productivity (*P_L_*) were estimated (Table 1, Table 2 and Table 3, Figure 1, Figure 2 and Figure 3). It was interesting to note that the microalgal cells displayed lipid accumulation in an irregular manner, regardless of their biomass formation. The highest lipid content was obtained in the heterotrophic culture with the increased feed glycerol concentration and was equivalent to 9.8%. Moreover, *P_L_* values under these conditions were enhanced. These results are in line with those reported from Heredia-Arroyo et al. [5] about *A. protothecoides*, according to which, even though the cell growth rate was negatively affected at higher glucose concentrations, the lipid contents were high (around 23%). It seems that, in such cases, glycerol, which is present in many lipids (glycerines), is assimilated into the lipid biosynthesis pathway and, thus, enhances the total lipid content. Comparing our results for the lipid content to the ones obtained for the proteins fraction under heterotrophic conditions (see Section 3.2.2), it is indicated that *A. protothecoides* assimilates glycerol firstly for synthesizing proteins and, if glycerol is in excess, then it directs it through lipids biosynthesis pathways. A similar observation was reported for *A. platensis* growing on glycerol [41]. 

Lipid content and lipid productivity decreased in general, with increasing dilution rates. The phenomenon of lipid content change as a function of the dilution rate was observed for several oleaginous microorganisms, including *C. vulgaris* and *Chlorella*
*pyrenoidosa* [10,11]. It seems that the microalgal cells need to remain in the reactor for an adequate time, so as to assimilate the available nitrogen and then convert the remaining carbon source to oil. In batch cultures, it has been established that lipid synthesis is only partially linked to microbial growth and occurs during the late exponential phase and the stationary (mixed growth associated) [44]. The same trend was also observed for carbohydrates, which also comprise the storage material of the microalgal cell (see Section 3.2.1). Furthermore, the increased protein contents displayed at higher *D* (Section 3.2.2) strengthen the assumption that, at these *D*, a shift in the metabolic pathways of glycerol assimilation occurs, leading carbon to the biosynthesis of free lipid products, such as amino acids.

The content of biomass lipids was significantly influenced by the presence of light in the cultivation system. Decreased lipids contents and decreased lipid productivities of the system were recorded under mixotrophic conditions. Similar results have been reported for *A. protothecoides* grown in glycerol under different peptone concentrations and light sources [14]. As is known, the expression of genes involved in chlorophyll metabolism, photosynthesis, and carotenoid biosynthesis is dramatically downregulated during heterotrophic microalgal growth, while genes for glycolysis, the TCA cycle and fatty acid synthesis are upregulated [45]. Under mixotrophic conditions, pigments production and photosynthesis are processes that commonly take place in the microalgal cells.

Although *A. protothecoides* has been reported in the literature to be capable of accumulating up to 70% of neutral lipids in its cellular compartments [5], in our study the lipid contents estimated were significantly lower. This dissimilar oil accumulation behavior may be attributed to the cultural conditions. There are many nutritional and environmental factors affecting lipid accumulation in microalgal cells, among which are the nitrogen source and its level in the growth medium [5]. In our study, the nitrogen levels perhaps were too high or the choice of peptone against yeast extract (which is more commonly used) was not so effective to cause the metabolic change. However, there are published data arguing that the accumulation of lipids may not be only due to nutrients exhaustion, but also the result of the carbon excess in the growth medium [46,47], a condition that applied to our study. Generally, it seems that, in our case, there was lack of a stress factor capable to inhibit cell growth while increasing the lipid content.

Several microalgal species are well known for producing specific classes of fatty acids in their cellular compartments through simple manipulatory treatments during their cultivation. In *A. protothecoides* cells, triacylglycerols are usually synthesized ranging from C_14_ to C_28_ when cultivated heterotrophically [8]. To define the impacts of different trophic modes on lipids constitution, fatty acid profiling was performed in the microalgal cells when grown in a continuous system with 30 g/L feed glycerol concentration at various *D*, under heterotrophic and mixotrophic conditions (Figure 5). 

Fourteen different types of fatty acids, including saturated, monounsaturated, and polyunsaturated, were detected in the heterotrophic mode culture. Oleic acid (C18:1 up to 55.6%), linoleic (C18:2 up to 19.8%), and palmitic acid (C16:0 up to 12.5%) were the major components of all the three oil categories and contributed more than approximately 85% to the total fatty acids. Similar results for lipid biosynthesis were observed by Rismani-Yazdi et al. [48] during the cultivation of *A. protothecoides* in a heterotrophic semicontinuous system in glucose. In the present study, there were no significant differences in the fatty acids profiles between the two trophic modes applied. Namely, under mixotrophic conditions, the percentages of the major fatty acids produced were: 61.2% for oleic, 18.8% for linoleic, and 11.4% for palmitic acid. Rismani-Yazdi et al. [48], in contrast, observed different fatty acid compositions between photoautotrophically and heterotrophically grown cells of *A. protothecoides.* This observation, in association with our results, may comprise an indication that *A. protothecoides*, in the simultaneous presence of light and carbon sources, prefers to act as a heterotrophic organism. The majority of the fatty acids detected in the present study were monounsaturated fatty acids. Methyl esters of monounsaturated fatty acids are considered to be a good source for biodiesel production, in comparison to polyunsaturated fatty acids. 

## 4. Conclusions

The continuous culture system was an effective method to study the growth kinetics, substrate assimilation, and high value metabolites production of *A. protothecoides* on different concentrations of crude glycerol (30 g/L and 50 g/L), under heterotrophic or mixotrophic conditions. Increased initial glycerol concentration in the heterotrophic culture provoked decreased biomass productivity and higher amounts of unconsumed glycerol, acting as a growth restrictive factor. *A. protothecoides*, unlike other *Chlorella* species, displayed better growth parameters under heterotrophic conditions compared with the mixotrophic ones, as was determined by the higher *μ_max_* and *P_max_* and the lower *K_S_* values, which mean that it can grow better and faster due to the more efficient exploitation of the substrate, resulting in a more effective system function. It, therefore, seems that *A. protothecoides* in the simultaneous presence of light and carbon sources performs better as a heterotrophic organism. Heterotrophy also had a positive impact on the lipids, proteins, and carbohydrates contents in the microalgal cells, but no significant effect on the cellular fatty and amino acids profiles, and, thus, the lipids and proteins composition. 

## Figures and Tables

**Figure 1 microorganisms-10-00541-f001:**
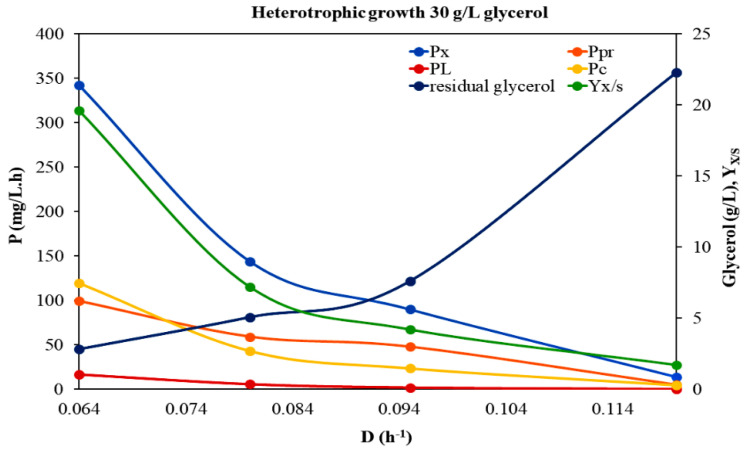
Effect of dilution rate (*D*) on the growth of *Auxenochlorella protothecoides* under heterotrophic conditions in BBM modified medium with initial glycerol concentration of 30 g/L. Τhe variation in biomass productivity (*P_x_*, mg/L·h), lipids productivity (*P_L_*, mg/L·h), proteins productivity (*P_Pr_*, mg/L·h), carbohydrates productivity (*P_c_*, mg/L·h) residual glycerol and yield coefficient on glycerol (*Y_x/S_*), with the dilution rate, are displayed. Τhe *Y_x/S_* has been increased a hundredfold (100 g/g).

**Figure 2 microorganisms-10-00541-f002:**
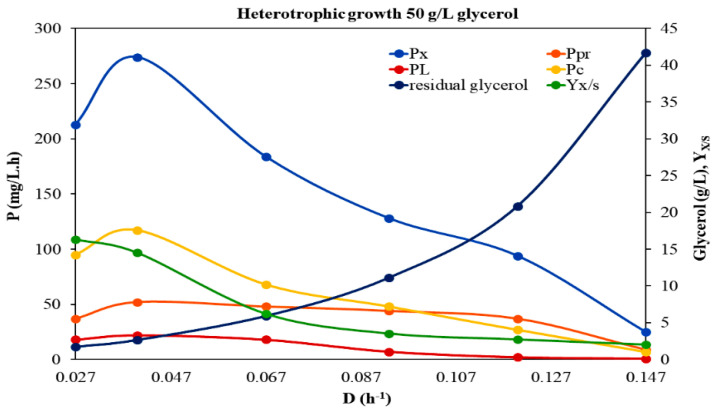
Effect of *D* on the growth of *A. protothecoides* under heterotrophic conditions in BBM modified medium with initial glycerol concentration of 50 g/L. Τhe variation in biomass productivity *P_x_* (mg/L·h), lipids productivity *P_L_* (mg/L·h), proteins productivity *P_Pr_* (mg/L·h), carbohydrates productivity *P_c_* (mg/L·h) residual glycerol and *Y_x/S_*, with *D*, are displayed. Τhe *Y_x/S_* has been increased a hundredfold (100 g/g).

**Figure 3 microorganisms-10-00541-f003:**
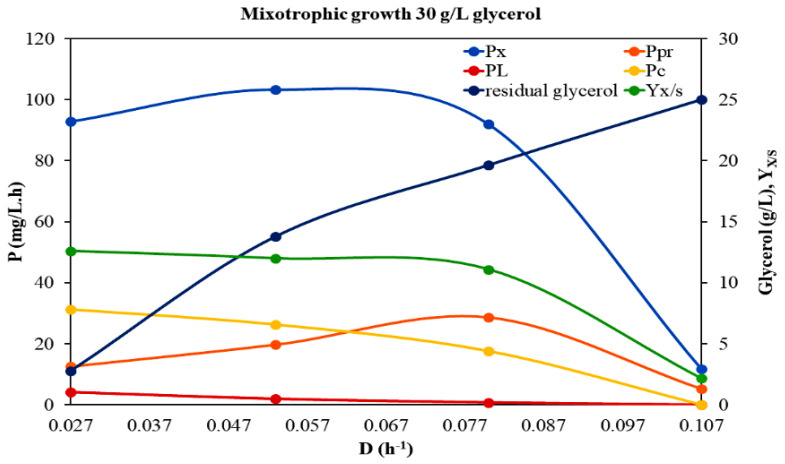
Effect of *D* on the growth of *A. protothecoides* under mixotrophic conditions in BBM modified medium with initial glycerol concentration of 30 g/L. Τhe variation in biomass productivity *P_x_* (mg/L·h), lipids productivity *P_L_* (mg/L·h), proteins productivity *P_Pr_* (mg/L·h), carbohydrates productivity *P_c_* (mg/L·h) residual glycerol and *Y_x/S_*, with *D* are displayed. Τhe *Y_x/S_* has been increased a hundredfold (100 g/g).

**Figure 4 microorganisms-10-00541-f004:**
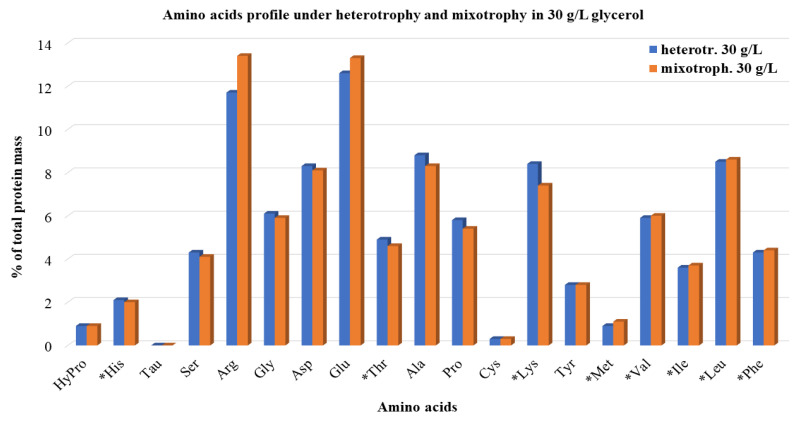
Amino acids profile analysis of *A. protothecoides* grown in 30 g/L initial crude glycerol concentration under heterotrophic and mixotrophic conditions (red monochromatic light, photoperiod of 16 h light/8 h dark). Values are the average of the percentages at all *D* applied, expressed as a % fraction of the total amount of proteins. The essential amino acids for human nutrition are indicated by an asterisk.

**Figure 5 microorganisms-10-00541-f005:**
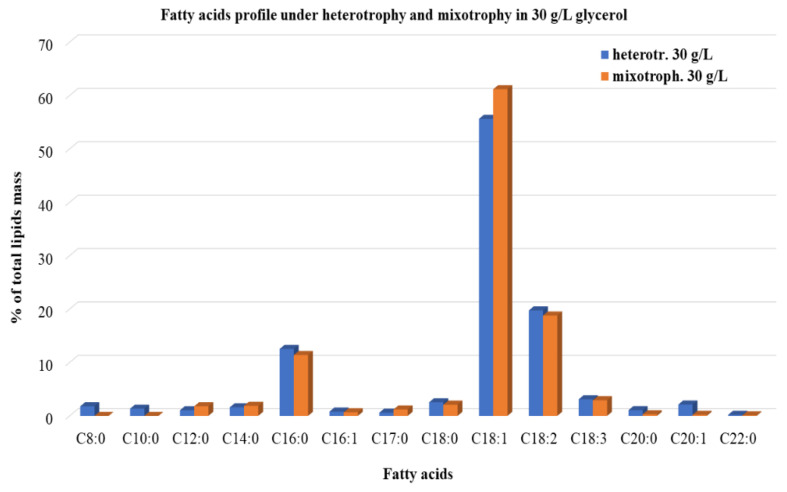
Fatty acids profile of *A. protothecoides* grown on 30 g/L initial crude glycerol concentration under heterotrophic and mixotrophic conditions (red monochromatic light, photoperiod of 16 h light/8 h dark). Values are the average of the percentages at all *D* applied, expressed as a % fraction of the total amount of lipids.

**Table 1 microorganisms-10-00541-t001:** Values of biomass, protein, lipids, and carbohydrates production (means ± SD) in a continuous flow culture of *Auxenochlorella protothecoides* with 30 g/L glycerol under heterotrophic conditions at steady state situations at various dilution rates (*D*). The productivities of biomass (*P_x_*) and metabolites (*P_Pr_* for proteins, *P_L_* for lipids, and *P_c_* for carbohydrates) and yield coefficient on glycerol (*Y_x/S_*) were calculated by the equations: *P_x_* = *D*·*x*, *P_m_* = *D*·*m* and *Y_x/S_* = *x/S_asimilated_*, respectively (where *S* is the substrate -glycerol).

Heterotrophic Growth 30 g/L Glycerol												
*D* (h^−1^)	Biomass (x) ^1^ (d.w. g/L)	Proteins (g/L)	Proteins ^2^ (% d.w.)	Lipids (g/L)	Lipids ^2^ (% d.w.)	Carbohydrates (g/L)	Carboh.^2^ (% d.w.)	*Y_x/S_*	*P_x_* (mg/L·h)	*P_Pr_* (mg/L·h)	*P_L_* (mg/L·h)	*P_c_* (mg/L·h)
0.064	5.341 ± 0.689	1.549 ± 0.175	29	0.2560 ± 0.0478	4.8	1.8694 ± 0.137	35	0.196	342	99.14	16.38	119.64
0.080	1.800 ± 0.236	0.738 ± 0.159	41	0.0710 ± 0.0210	3.9	0.5400 ± 0.080	30	0.072	144	59.04	5.62	43.2
0.095	0.950 ± 0.122	0.500 ± 0.089	53	0.0193 ± 0.0023	2.0	0.2470 ± 0.061	26	0.042	90	47.7	1.80	23.47
0.120	0.130 ± 0.022	0.046 ± 0.013	35	0.0010 ± 0.0006	0.8	0.0377 ± 0.014	29	0.017	14	4.9	0.10	4.52

^1^ dry weight (g/L), ^2^ percentage % of biomass.

**Table 2 microorganisms-10-00541-t002:** Values of biomass, protein, lipids, and carbohydrates production (means ± SD) in a continuous flow culture of *A. protothecoides* with 50 g/L glycerol under heterotrophic conditions at steady state situations at various *D*. The productivities of biomass (*P_x_*) and metabolites (*P_Pr_* for proteins, *P_L_* for lipids, and *P_c_* for carbohydrates) and *Y_x/S_* were calculated by the equations: *P_x_* = *D*·*x*, *P_m_* = *D*·*m* and *Y_x/S_* = *x/S_asimilated_*, respectively.

Heterotrophic Growth 50 g/L Glycerol												
*D* (h^−1^)	Biomass (×) ^1^ (d.w. g/L)	Proteins (g/L)	Proteins ^2^ (% d.w.)	Lipids (g/L)	Lipids ^2^ (% d.w.)	Carbohydrates (g/L)	Carboh.^2^ (% d.w.)	*Y_x/S_*	*P_x_* (mg/L·h)	*P_Pr_* (mg/L·h)	*P_L_* (mg/L·h)	*P_c_* (mg/L·h)
0.027	7.888 ± 1.072	1.36 ± 0.13	17.2	0.670 ± 0.088	3.5	3.518 ± 0.206	44.6	0.163	212.98	36.72	18.09	94.99
0.040	6.850 ± 1.168	1.31 ± 0.16	19.1	0.548 ± 0.077	4.5	2.925 ± 0.241	42.7	0.144	274.00	52.40	21.92	117.00
0.067	2.750 ± 0.923	0.72 ± 0.15	26.2	0.270 ± 0.042	9.8	1.015 ± 0.175	36.9	0.062	184.25	48.24	18.09	68.00
0.093	1.380 ± 0.259	0.47 ± 0.12	34.4	0.080 ± 0.016	5.8	0.520 ± 0.147	37.7	0.035	128.34	43.71	7.44	48.36
0.120	0.780 ± 0.085	0.31 ± 0.07	40.0	0.017 ± 0.005	2.2	0.229 ± 0.041	29.4	0.027	93.60	37.20	2.04	27.48
0.147	0.167 ± 0.020	0.06 ± 0.01	35.0	0.004 ± 0.001	2.4	0.050± 0.013	30.2	0.020	24.55	8.82	0.59	7.35

^1^ dry weight (g/L), ^2^ percentage % of biomass.

**Table 3 microorganisms-10-00541-t003:** Values of biomass, protein, lipids and carbohydrates production (means ± SD) in a continuous flow culture of *A. protothecoides* with 30 g/L glycerol under mixotrophic conditions at steady state situations atn various *D*. The productivities of biomass (*P_x_*) and metabolites (*P_Pr_* for proteins, *P_L_* for lipids, and *P_c_* for carbohydrates) and *Y_x/S_* were calculated by the equations: *P_x_* = *D*·*x*, *P_m_* = *D*·*m* and *Y_x/S_* = *x/S_asimilated_*, respectively.

Mixotrophic Growth 30 g/L glycerol												
*D* (h^−1^)	Biomass (×) ^1^ (d.w. g/L)	Proteins (g/L)	Proteins ^2^ (% d.w.)	Lipids (g/L)	Lipids ^2^ (% d.w.)	Carbohydrates (g/L)	Carboh ^2^ (% d.w.)	*Y_x/S_*	*P_x_* (mg/L·h)	*P_Pr_* (mg/L·h)	*P_L_* (mg/L·h)	*P_c_* (mg/L·h)
0.027	3.44 ± 0.397	0.46440 ± 0.0352	13.50	0.15136 ± 0.0367	4.40	1.162 ± 0.158	33.78	0.126	92.88	12.54	4.09	31.37
0.053	1.95 ± 0.132	0.37226 ± 0.02665	19.09	0.03705 ± 0.01078	1.90	0.498 ± 0.173	25.54	0.120	103.35	19.73	1.96	26.39
0.080	1.15 ± 0.229	0.35892 ± 0.02692	31.21	0.00978 ± 0.00238	0.85	0.221 ± 0.033	19.26	0.111	92.00	28.71	0.78	17.68
0.107	0.11 ± 0.035	0.04890 ± 0.00593	44.45	0.00018 ± 0.00005	0.16	0	0	0.022	11.77	5.23	0.019	0

^1^ dry weight (g/L), ^2^ percentage % of biomass.

## Data Availability

The data presented in this study are available in the article.

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
