# Peer review of "Continuous Culture of Auxenochlorella protothecoides on Biodiesel Derived Glycerol under Mixotrophic and Heterotrophic Conditions: Growth Parameters and Biochemical Composition"

_microorganisms, 2022, doi:10.3390/microorganisms10030541_

Round 1
Reviewer 1 Report
It is a good and interesting paper. I have two remarks. Introduction section should be extended. The second remark is Authors must put in conclusion section.
Author Response
Response to Reviewer 1 Comments
Point 1: Introduction section should be extended.
Response 1: Introduction was extended according to the Reviewer’s suggestion
Point 2: Authors must put in conclusion section.
Response 2: A Conclusion section was added following the Reviewer’s suggestion
Reviewer 2 Report
Reviewer’s comments
The manuscript titled “Continuous culture of Auxenochlorella protothecoides on biodiesel-derived glycerol under mixotrophic and heterotrophic conditions: Growth parameters and biochemical composition” is important due to the fact that it discusses the growth of Auxenochlorella protothecoides under heterotrophic and mixotrophic conditions. Its growth kinetics were evaluated under steady-state conditions. This manuscript also summarizes information about the algal biomass that contains many essential and non-essential amino acids especially arginine, glutamic acid, lysine, aspartic acid, leucine, and alanine.
The comments on particular parts of the text are to be found below.
Abstract
In the Abstract there is no information about microalga of Auxenochlorella protothecoides and their growth parameters and biochemical composition (except for one sentence). Please rewrite the abstract. It would be clearer if the authors concentrated on main problems and removed the suggestions.
Introduction
The introduction briefly summarizes knowledge about biodiesel and glycerol as primary by-product. The Authors described microalgae as fast-growing and unicellular species widely distributed in nature, that stand out for their high-value biomass. However, there is nothing about the ecology and morphology of main species Auxenochlorella protothecoides (Line 79-81, Line 90-92 lack references). Please introduce more information about habitat, distribution, biology and ecology of study species and source of biofuel production. The high lipid content of this alga during heterotrophic growth is promising for biodiesel. Do the Authors have photographic documentation of A. protothecoides?
Materials and methods
I am positively surprised by the very detailed description of the methods used.
Author Response
Response to Reviewer 2 Comments
Point 1: In the Abstract there is no information about microalga of Auxenochlorella protothecoides and their growth parameters and biochemical composition (except for one sentence). Please rewrite the abstract. It would be clearer if the authors concentrated on main problems and removed the suggestions.
Response 1: The authors rewrote the abstract according to the Reviewer’s suggestion. Information about the growth parameters and the microalgal biochemical composition were added, while any Authors’ suggestions were removed.
Point 2: The introduction briefly summarizes knowledge about biodiesel and glycerol as primary by-product. The Authors described microalgae as fast-growing and unicellular species widely distributed in nature, that stand out for their high-value biomass. However, there is nothing about the ecology and morphology of main species Auxenochlorella protothecoides (Line 79-81, Line 90-92 lack references). Please introduce more information about habitat, distribution, biology and ecology of study species and source of biofuel production. The high lipid content of this alga during heterotrophic growth is promising for biodiesel. Do the Authors have photographic documentation of A. protothecoides?
Response 2: The Introduction was revised following the suggestions of the Reviewer. We added references to Lines 79-81 and 90-92 and the relating to the manuscript information about the ecology, biology, morphology and the potential of employing Auxenochlorella protothecoides in the production of biodiesel. The Authors have photographic documentation from the microscopic observations of the microalgal cells, though no significant differences compared to the typical cell of A. protothecoides that has already been described in the literature were recorded, and hence, do not worth reporting.